# Artichoke By-Products Valorization for Phenols-Enriched Fresh Egg Pasta: A Sustainable Food Design Project

**Tiziana Amoriello** [1,*] **, Francesco Mellara** [1] **, Stefania Ruggeri** [1] **, Roberto Ciorba** [2] **, Danilo Ceccarelli** [2] **and Roberto Ciccoritti** [2,*]

[1] CREA Research Centre for Food and Nutrition, Via Ardeatina 546, 00178 Rome, Italy
[2] CREA Research Centre for Olive, Citrus and Tree Fruit, Via di Fioranello 52, 00134 Rome, Italy
* Correspondence: tiziana.amoriello@crea.gov.it (T.A.); roberto.ciccoritti@crea.gov.it (R.C.)

**Abstract:** More healthy and sustainable food are nowadays desirable to improve human health and protect the planet's resources. From this perspective, the aim of this study is to investigate artichoke (*Cynara scolymus* L.) by-products as a potential source of phenolic compounds and to use these compounds to design new fresh egg pasta formulation. Sustainable extraction was carried out using ultrasound-assisted extraction (UAE) and chemometric techniques, such as the Response Surface Methodology (RSM). UAE process parameters (temperature and time) and solvent composition (ethanol aqueous mixtures) were optimized using a three-level Box–Behnken design, in order to carry out the maximum yield in phenols. Under the optimal conditions (temperature: 60 °C; time: 60 min; solvent: 50% ethanol:water), the amount of phenolics (TPC) was $22.4 \pm 0.2$ mg GAE g$^{-1}$ d.w., characterized mainly by dicaffeoylquinic acid ($32.8 \pm 0.6$ mg CAE g$^{-1}$ d.w.) and chlorogenic acid ($14.1 \pm 0.2$ mg CAE g$^{-1}$ d.w.). Hence, the polyphenols extract was used as an ingredient to design a new formulation of functional fresh egg pasta. Four recipes with soft wheat and semolina ($P_1$ and $P_2$) and with soft wheat alone ($P_3$ and $P_4$) were prepared. Compared with control pastas ($P_1$ and $P_3$), the enriched ones ($P_2$ and $P_4$) showed a higher polyphenol content, especially for $P_4$ ($1.86 \pm 0.03$ mg GAE g$^{-1}$ d.w. for $P_1$, $2.05 \pm 0.02$ mg GAE g$^{-1}$ d.w. for $P_2$, $1.92 \pm 0.03$ mg GAE g$^{-1}$ d.w. for $P_3$, $2.04 \pm 0.02$ mg GAE g$^{-1}$ d.w. for $P_4$). A high decrease in TPC was observed as a result of the cooking process, especially for the two control formulations ($-71\%$ for $P_1$ and $-70\%$ for $P_3$) in comparison with $P_2$ ($-64\%$) and $P_4$ ($-55\%$). At last, to assess the antimicrobial effect of artichoke by-products on fresh pasta and to monitor its spoilage, we used image analysis. Corresponding to a higher TPC content, $P_2$ and $P_4$ showed an extended shelf life of 16% and 33%, respectively, probably due to the antioxidant activity of artichoke. The new fresh egg pasta enriched with polyphenols extracted from artichoke by-products showed very good nutritional and technological characteristics, even after cooking, confirming the good potentiality of artichoke by-products in the design of new, healthy, and sustainable food products.

**Keywords:** artichoke; by-products; phenolic compounds; green extraction; food design; Response Surface Methodology; fresh egg pasta

## 1. Introduction

International strategies, such as the 2030 Agenda for Sustainable Development, the FAO Strategic Framework 2022-31, and the European Green Deal, strive to achieve a more sustainable future through a radical shift of the agri-food systems towards more efficient, inclusive, resilient, and sustainable ones to accomplish better production, nutrition, environment, and ultimately a better life [1–3]. Achieving these goals requires deep evaluation of the agri-food systems, upon circular economy principles, to speed up the transformative processes of these systems from a sustainability perspective [4–6]. From this new standpoint, the transition to re-designed food systems has to ease hunger issues, supply required nutrients, prevent nutrition-related diseases, and enhance consumers' physical

and mental health [7]. The recovery and reuse of the fruits and vegetables by-products, with the aim to extract functional compounds and to develop innovative food products is a good approach to circularity also from a food–health relationship perspective, due to their considerable amount of valuable compounds (fiber, minerals, vitamins, and bioactives) [8]. Food industries view functional foods with ingredients from natural and cheap sources, i.e., agri-food residues, as interesting economic commodities [9].

Among all the by-products of the numerous edible plant species with functional properties, those of the *Cynara scolymus* L. are considered by food industries as promising sources. Globe artichoke (*Cynara scolymus* L.) is a perennial herbaceous plant belonging to the Asteraceae family and is most likely native to the Mediterranean area. In 2019, global artichoke production and harvested areas were gauged to be 1,594,385 tons and 122,644 ha, respectively [10]. Artichokes are mainly cultivated in the Mediterranean area. Italy, Egypt, and Spain are the world's top three artichoke producers, with a production of 378,820 tons (23.8% of the world production), 296,899 tons (18.6% of the world production), and 199,940 tons (12.5% of the world production), respectively. The edible portion of the plant consists of the immature fleshy leaves (bracts) and part of the stem just below it, named capitulum or head [11]. It is highly appreciated by consumers for its pleasant, bitter taste due to cynarin [12], and its nutritional value, being rich in inulin, fibers, minerals, and bioactive compounds, including polyphenols such as cynarin, caffeoylquinic, chlorogenic acid, and flavonoids such as luteolin or its glycosides [13,14]. Furthermore, artichoke's consumption provides significant health benefits, due to the hepato-protective, cardio-protective, anti-inflammatory, and antioxidative activities, as well as the ability to inhibit cholesterol biosynthesis and low-density lipoprotein oxidation [15–17].

The edible portion of globe artichoke is around 15–25% of its fresh weight, while stems, leaves, and external bracts (the remaining 75–85%) are discarded by the artichoke processing industry. This processing generates large amounts of by-products that could negatively impact the environment. However, these biowastes can be successfully used for different purposes. They can be employed for animal feed or as manure [18]; as a vegetable alternative and milk coagulants for the manufacture of dairy products [19]; as ingredients for baked products and jam [19]; as gelling agents in food application or prebiotics for healthy gut flora [20–22]; as a source of polysaccharides, mainly cellulose and hemicellulose, for bioenergy production [19]; or for usage in paper pulp production [23] and poly-lactic acid (PLA)-based biocomposites [24]. An economically viable solution for the valorization of these by-products is their recovery for pharmaceutical or cosmetic purposes.

In this sight, the valorization of artichoke residues has a potential impact on the sustainability [25] of the food processing industry and is an opportunity to contribute to the sustainable design of new products in the functional pasta production area, meeting the needs from both an economic and social point of view. Processing and manufacturing conditions, product design, and formulation have to be carefully considered because their lack of optimization could greatly influence the successful development of novel products [26]. Recently, polyphenols from agri-food residues were extracted using a green technology as ultrasounds, being an efficient, environmentally friendly, sustainable, and cost-effective tool for bioactive compound extraction in accordance with the principles of Green Chemistry [27,28]. Compared with other extraction techniques (conventional, supercritical fluid extraction, microwave-assisted extraction, saponification, enzymatic hydrolysis, pressurized liquid extraction, liquid–solid extraction, etc.), ultrasound-assisted extraction (UAE) is up-and-coming for the extraction of thermolabile phenolic compounds also due to the reduced solvent use and energy consumption [27]. In order to do this, extraction process parameters such as temperature, time, and solvent composition should be optimized to lead to a maximum yield in phenols.

Upon these considerations, the aim of this study is to investigate the potentiality of ultrasound-assisted extraction as a green extraction technique for the recovery of phenolic compounds from globe artichoke by-products (e.g., bracts and stems) and design a functional fresh egg pasta, a traditional Italian product very popular worldwide. Furthermore,

to the best of our knowledge, there are few studies that have investigated green extraction technologies to extract bioactive compounds from agri-food by-products for their addition to pasta, and there is only one study on the enrichment of fresh pasta (Orecchiette) with artichoke by-products extract with this purpose [29]. The novelty of our study concerns the identification of optimal process conditions for the UAE extraction of phenolic compounds from artichoke residues using Response Surface Methodology. On the contrary, Pasqualone et al. [29] set extraction parameters at fixed levels. The optimal phenolic extract was used to enrich two different types of fresh egg pasta (Fettuccine). Differences in phenolic content in fresh egg pasta, with and without enrichment, and before and after cooking, were tested in order to estimate polyphenol cooking loss. Ultimately, it was assessed using image analysis whether the artichoke extract positively influenced the pasta shelf life.

## 2. Materials and Methods

### 2.1. Raw Materials

Artichoke by-products were obtained by waste processing of artichokes (cultivar "Campagnano"); immediately placed in the refrigerator at 5 °C; and about eight hours after, oven-dried at 60 °C for 24 h until the moisture content was less than 10%, below which microbial growth is strongly inhibited [27]. The dried artichoke by-products (e.g., mixture of bracts and stems) were grounded with a Bühler MLI 203 sifter (Milan, Italy) and sieved to obtain a fine flour with particles sized from 400 to 500 μm.

Commercial soft wheat flour (Molini PROGEO SCA, Masone, Italy), durum wheat semolina (F.lli De Cecco di Filippo Fara S. Martino S.p.A., Fara S. Martino, Italy), and eggs (Gruppo Novelli srl, Terni, Italy) were purchased from a local market. The protein contents of soft wheat flour, semolina, and eggs, reported on the package of the products, were 12.5 g/100 g, 14.0 g/100 g, and 13.0 g/100 g, respectively.

### 2.2. Proximate Composition

Moisture, proteins, and ashes were determined by the ICC standard methods 110/1, 105/2, and 104/1, respectively [30]. Protein content was estimated using the conversion factor 6.25 for artichoke flours. Total dietary fiber (TDF) content was measured according to Lee et al. [31], using a reagent kit (K-TDFR, Megazyme Int., Wicklow, Ireland). All determinations were made in triplicate and the result expressed as dry weight (d.w.).

### 2.3. FTIR Analysis

An FTIR-ATR spectrometer (Nicolet iS 10 FT-IR Thermo Fisher Scientific Inc., Waltham, MA, USA) equipped with a diamond crystal cell (ATR) was used for MIR spectra acquisition. The spectra were acquired in the wavenumber region 650–4000 cm$^{-1}$, as described by Amoriello et al. [32], and then processed with the OMNIC™ software (Thermo Fisher Scientific Inc., Waltham, MA, USA).

### 2.4. Extraction of Phenolic Compounds by Ultrasound-Assisted Extraction and Optimization Procedure

Phenolic compounds in artichoke by-products were extracted with ultrasound-assisted extraction (UAE) by an ultrasonic bath (ElmasonicS30H, Elma Ultrasonic Technology, Singen, Germany). For each extraction batch, 2 g of dried and milled artichoke by-products was weighted in a tube and diluted with 40 mL of different solvents mixed and sonicated at 37 kHz for different times and at a heating power of 200 W, as described in Table 1. The solvent used for the extraction was a mixture of ethanol and water (Table 1).

**Table 1.** Box–Behnken design with coded and uncoded parameters of ultrasound-assisted extraction (UAE), and experimental and predicted response values (total phenolic content, TPC) in the extracts using Response Surface Methodology (RSM).

| Run | $X_1$ | $X_2$ | $X_3$ | $X_1$ (%) | $X_2$ (min) | $X_3$ (°C) | TPC$_{exp}$ (mg GAE/g d.w.) | TPC$_{RSM}$ (mg GAE/g d.w.) |
|---|---|---|---|---|---|---|---|---|
| 1 | −1 | −1 | 0 | 40 | 20 | 50 | 18.26 ± 0.04 | 18.26 |
| 2 | 1 | −1 | 0 | 80 | 20 | 50 | 13.79 ± 0.05 | 13.75 |
| 3 | −1 | 1 | 0 | 40 | 60 | 50 | 19.43 ± 0.05 | 19.48 |
| 4 | 1 | 1 | 0 | 80 | 60 | 50 | 13.55 ± 0.07 | 13.55 |
| 5 | −1 | 0 | −1 | 40 | 40 | 40 | 20.07 ± 0.02 | 20.06 |
| 6 | 1 | 0 | −1 | 80 | 40 | 40 | 14.24 ± 0.04 | 14.29 |
| 7 | −1 | 0 | 1 | 40 | 40 | 60 | 22.27 ± 0.04 | 22.22 |
| 8 | 1 | 0 | 1 | 80 | 40 | 60 | 17.54 ± 0.02 | 17.55 |
| 9 | 0 | −1 | −1 | 60 | 20 | 40 | 18.19 ± 0.04 | 18.19 |
| 10 | 0 | 1 | −1 | 60 | 60 | 40 | 16.92 ± 0.08 | 16.88 |
| 11 | 0 | −1 | 1 | 60 | 20 | 60 | 19.04 ± 0.03 | 19.08 |
| 12 | 0 | 1 | 1 | 60 | 60 | 60 | 21.41 ± 0.02 | 21.42 |
| 13 | 0 | 0 | 0 | 60 | 40 | 50 | 18.22 ± 0.02 | 18.22 |
| 14 | 0 | 0 | 0 | 60 | 40 | 50 | 18.22 ± 0.02 | 18.22 |
| 15 | 0 | 0 | 0 | 60 | 40 | 50 | 18.22 ± 0.02 | 18.22 |

$X_1$ = solvent composition (%); $X_2$ = extraction time (min); $X_3$ = extraction temperature (°C).

A Box–Behnken design (BBD) with three independent factors (solvent composition, extraction time, and extraction temperature), combined with multiple responses and desirability functions, was applied for extraction optimization. The design included 15 runs (three at the central point); each run was replicated three times. For three factors, the Box-Behnken design offered some advantage in requiring a fewer number of runs in comparison with other designs—in particular, the Central Composite Designs (CCC-circumscribed, CCF-face centered, and CCI-inscribed). Each factor was coded at three levels, −1, 0, and +1 (Table 1). Solvent composition ($X_1$) is a mixture of water and ethanol; ethanol percentage ranged between 40% and 80%. Extraction time ($X_2$) varied between 20 and 60 min, whereas extraction temperature ($X_3$) ranged from 40 to 60 °C. These values were assessed by the literature [33–36] and resulted from a pilot study (data not shown). Times and temperatures up to 60 min and 60 °C were tested because the pilot study demonstrated that higher extraction times and temperatures increased the yields by only about 5% and were not economically sustainable.

Powdered artichoke residues were dissolved in the extraction solvent and the mixture was heated and sonicated for a time as in Table 1. Water–ethanol mixture ratio 20:80 (*v/v*) was fixed in this study. At the end of sonication, the extracted product was left to cool at room temperature, then centrifuged at 3000 rpm for 20 min at 4 °C. At last, each sample was filtered with a nylon syringe filter (0.45 μm) and the supernatants were immediately analyzed.

The predicted response of TPC (TPC$_{RSM}$) was obtained using a second-order polynomial equation, as follows:

$$Y_i = \beta_0 + \sum_{i=1}^{3} \beta_i X_i + \sum_{i=1}^{3} \beta_{ii} X_{ii}^2 + \sum_{ij,\ i<j} \beta_{ij} X_i X_j + e_i \qquad (1)$$

where Y = total phenolic content (TPC); $X_1$ = solvent composition (%); $X_2$ = extraction time (min); $X_3$ = extraction temperature (°C); $\beta_0$ = intercept; $\beta_i$, $\beta_{ii}$, $\beta_{ij}$ = linear, quadratic, and interactive coefficients, respectively; $e_i$ = error term. Statistical significance and goodness of fit of applied model, and optimization procedure with 3D plots and desirability functions were assessed following the method described by Iadecola et al. [27].

Under optimal conditions, the extraction was repeated; the final extract was subjected to a rotary evaporator (Büchi® Rotavapor® R-124, Laguna Hills, CA, USA) at 40 °C and 175 mbar to reduce the presence of ethanol.

To evaluate the UA extraction efficiency, polyphenols from artichoke by-products samples were also extracted in the same conditions but without ultrasound. Briefly, 2 g of samples were mixed with 40 mL of extraction solution (50% ethanol:water) at 60 °C for 60 min. Afterwards, the suspension was cooled and filtered as previously reported for UA extraction.

### 2.5. Determination of Total Phenolic Content

The total phenolic content (TPC) of extracts was determined using the Folin–Ciocalteu (F–C) method as reported by Ciccoritti et al. [37]. TPC was calculated from a calibration curve, using Gallic acids as a standard. Results are expressed as milligrams of Gallic acid equivalents (GAE) per g of whole milled artichoke (d.w.).

### 2.6. Profiling of Phenolic Compounds by UHPLC-ESI-QTOF MS

Chromatographic separations were performed on an Agilent 1290 Infinity Binary LC with UV–Vis photodiode array detector (DAD G4212A), coupled with an Agilent 6530 Accurate Mass Quadrupole Time-of-Flight, equipped with a dual spray ESI source applying the following elution binary gradient at a flow rate of 0.35 mL min$^{-1}$; 0–10 min, isocratic 95% A (water/formic acid, 99.9/0.1 [*v/v*]), 5% B (acetonitrile/ formic acid, 99.9/0.1 [*v/v*]); 10–13 min, linear from 5–17% B; 13–18 min, isocratic 17% B; 18–25 min, linear from 17–45% B; 25–30 min, linear from 45–65% B; 30–34 min, linear from 65–90% B; 34–44 min, isocratic 90% B. Before the injection, the extract was diluted (1:5 *v/v*) with methanol water solution (80/20 *v/v* containing 2 mM sodium fluoride and hydrochloric acid (5 mM)), filtered through a 0.2 μm filter (Acrodisc®, Pall Corporation, New York, NY, USA), and 1 μL of filtered and diluted extract was injected (full loop injection) (1 μL). Kinetex C18 (150 × 2.1 mm i.d., 1.7 μm) from Phenomenex (Torrance, CA, USA), placed in a column oven set at 30 °C was used. Eluted compounds were detected using electrospray mass spectrometric analysis in negative mode and nitrogen was used as nebulizer gas at a pressure of 60 psi, flow of 8 L min$^{-1}$, and a temperature of 350 °C; capillary voltage was set to 2800 V and fragmentor voltage fixed at 175 V. All data were acquired in centroid mode.

UHPLC-ESI-QTOF MS files were processed using Mass Hunter Qualitative and Quantitative Analysis software (ver. B.05.00) to assist in adjacent peak deconvolution and background subtraction. Metabolites were characterized by their UV–Vis spectra (220–600 nm), retention times relative to external standards, mass spectra, and comparison to our in-house database, phytochemical dictionary of natural products database, and reference literature. More abundant phenols concentrations were expressed as mg g$^{-1}$ d.w. using a single standard of chlorogenic acid and coumaric acids (hydroxycinnamic acids), catechin (flavanols), and quercitin rutinoside (flavone). Standard calibration curves were constructed for each standard using four concentrations spanning 10, 25, 50, and 100 ppm. Assays were carried out in triplicate.

### 2.7. Design of Fresh Egg Pasta Enriched with Polyphenols from Artichoke By-Products

Four recipes of traditional Italian fresh egg pasta ($P_1$, $P_2$, $P_3$, $P_4$), i.d. Fettuccine, were prepared, as described in Table 2. $P_1$ and $P_3$—without the addition of the extract—were considered as "control" for pasta, with or without semolina, respectively. The percentages of flour, semolina, and egg ingredients were consistent with the traditional Fettuccine recipe. In particular, $P_1$ is the typical recipe for an industrial pasta, whereas $P_3$ is for a household product. The quantity of extract was considered at 10% as, at a higher level, it would have excessively and negatively altered the Fettuccine flavor and dough machinability. Water was added to the $P_1$ to helping gluten dough, as suggested by Balli et al. [38].

**Table 2.** Ingredients in percentage for the formulation of 4 different recipes for fresh egg pasta ($P_i$).

|                        | $P_1$ | $P_2$ | $P_3$ | $P_4$ |
|------------------------|-------|-------|-------|-------|
| Semolina (%)           | 35    | 35    | 0     | 0     |
| Soft wheat (%)         | 35    | 35    | 63    | 63    |
| Eggs (%)               | 20    | 20    | 37    | 27    |
| Water (%)              | 10    | 0     | 0     | 0     |
| Artichoke extract (%)  | 0     | 10    | 0     | 10    |

The ingredients were mixed for 4 min in a planetary mixer with accessories for Fettuccine (Kenwood KPL9000S, Hampshire, UK), until a homogeneous dough was obtained. Once the dough was formed, it was divided by hand in rectangular sizes of 30 cm and laminated to a final pasta thickness of 1 mm. Then, the pieces were cut using a roller-sheeter. A total 50 g of Fettuccine for each type of pasta were stored in a closed plastic vessel at 5 °C, a temperature generally used for storage of this product [39]. Another 50 g of pasta was cooked for 5 min in 0.75 L of boiling water.

*2.8. Color Measurement and Image Analysis*

Color measurements were taken on artichoke flours and fresh egg pasta using a Chroma Meter CR-200 (Konica Minolta, Tokyo, Japan) and (CIE) L*a*b* scale. The results (L*, a*, b*) are the average of measurements of five different points per sample.

To assess the spoilage of fresh pasta, usually related to mold growth [40], images of central pasta dish of each type of four pasta types were acquired twice using a Nikon D850 digital camera at a high resolution and a color depth of 16 bits, saving the images in uncompressed RAW format. A region of interest (ROI) of 650 × 650 pixels, representative of the whole sample surface, was extracted from each image using Adobe Photoshop. Then, the collected images were processed by Image Pro 10 software (Media Cybernetics Inc., Rockville, MD, USA) to monitor the presence of mold. In fact, mold spoilage is often visible once the colonies reach a diameter of 3 mm [39]. The pasta samples were observed daily for 8 days and considered "non-conforming" as soon as a colony appeared, according to Zardetto et al. [39]. The time employed by the mold colonies to reach a diameter of 3 mm is defined as "rejection time".

*2.9. Statistical Analysis*

All tests were replicated three times, and mean values and standard deviations were calculated. A one-way analysis of variance (ANOVA) employing the Kruskal–Wallis non-parametric test at a significance level of 5% was carried out to determine significant differences in all measured properties.

Generalized linear model (GzLM) was applied to assess the influence of artichoke by-products extract on pasta shelf life. Two factors (type of pasta and day) were considered. We also assumed the response variable Y had a Poisson distribution and the logarithm of its expected value E(Y) could be modeled by a linear combination of unknown parameters. Poisson regression models were GzLM with the logarithm as the canonical link function and the Poisson distribution function. The statistical model was

$$\eta = g(\mu) = \log(Y) = \beta_0 + \tau_i + \delta_k + \varepsilon_{ik}$$

where Y = number of mold colonies; $\beta_0$ = average effect common to all observations; $\tau_i$ = type of pasta (P); i = 1, . . . , 4; $\delta_k$ = day; $\varepsilon_{ik}$ = error term. The goodness of fit of GzLM model could be based on the deviance statistic, approximated by a chi-square distribution. We used the log-likelihood value to measure the goodness of fit of the regression models.

Data were processed using SPSS statistical software (version 22, SPSS, Chicago, IL, USA).

The experimental design, RSM analysis, and optimization procedure were carried out using Statistica statistical package software (Stat Soft Inc., Tulsa, OK, USA).

## 3. Results and Discussion

### 3.1. Artichoke By-Products Characterization

The moisture content, proteins, ashes, and total dietary fiber of artichoke by-products were $8.20 \pm 0.01$ g/100 g, $19.34 \pm 0.04$ g/100 g d.w., $10.31 \pm 0.05$ g/100 g d.w., and $51.8 \pm 0.4$ g/100 g d.w., respectively.

The mean FTIR spectrum of powdered artichoke by-products in the wavenumber region 650–4000 cm$^{-1}$ and its characteristic peaks are shown in Figure 1 and Table 3.

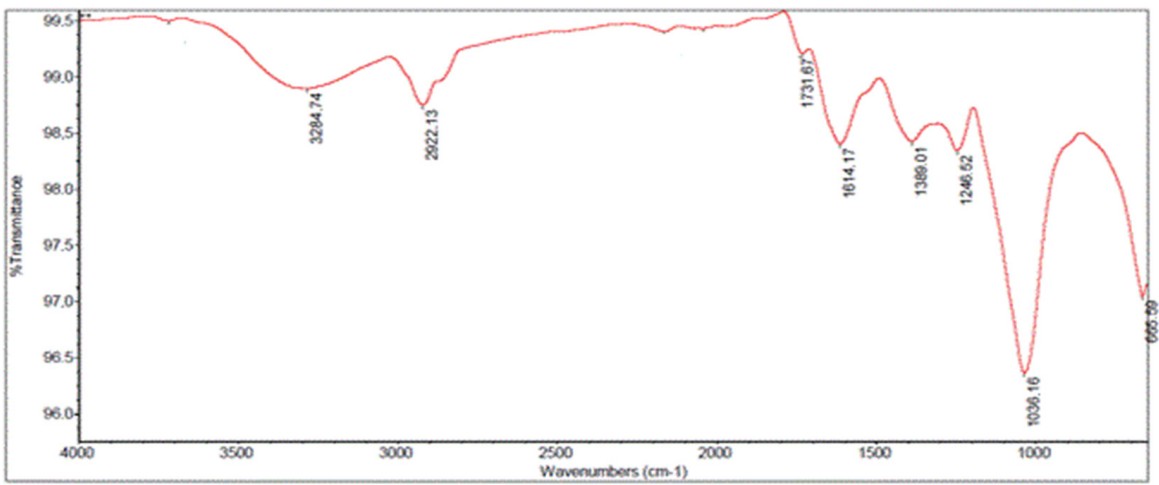

**Figure 1.** Mean FTIR spectrum of powdered artichoke by-products.

**Table 3.** Main features of mean FTIR spectrum of powdered artichoke by-products.

| Main Peak (cm$^{-1}$) | Typical Band | Compound Family |
|---|---|---|
| 3284.74 | O–H stretching vibration of the hydroxyl group | Polysaccharide compounds |
| 2922.13 | C–H stretching vibration | Polysaccharides such as inulin |
| 1731.67 | C=O stretching vibration carboxylic acid/ester groups | Lignin or hemicellulose |
| 1614.17 | C–C bonds of aromatic ring | Lignin or polyphenols |
| 1389.01 | CH$_2$ symmetric bending and COH deformation of phenols | Hemicelluloses, celluloses, and phenols |
| 1246.52 | OH group attached to an aromatic ring | S rings and G rings of lignin |
| 1036.16 | C–C stretching/C–O stretching | Polysaccharides such as inulin |
| 665.59 | O–H stretching vibration | Polysaccharides |

Peaks could be attributed mostly to polysaccharides with two characteristic broad absorption bands at 3500–3000 cm$^{-1}$ (O–H stretching vibration of the hydroxyl group) and 2900–2800 cm$^{-1}$ (C–H stretching vibration of methyl and methylene), as previously reported by Yan et al. [41]. In detail, the spectrum showed polysaccharides' characteristic peaks at 1036 cm$^{-1}$ (C–C stretching and C–O stretching), 2922 cm$^{-1}$ (C–H stretching vibration), and 3284 and 665 cm$^{-1}$ (O–H stretching vibration), in accordance with Quintero Ruiz et al. [42], who attributed the peaks of hemicellulose, lignin, and cellulose in the range 3000–1350 cm$^{-1}$ (C–H stretching, C=O stretching, and CH$_2$ symmetric bending). Moreover, the 1350–1700 cm$^{-1}$ region was assigned at phenolic compounds, as highlighted by Lavecchia et al. [43]. In particular, the peak at 1389 cm$^{-1}$ was attributed to COH deformation of phenols, as well as the peak at 1614 cm$^{-1}$, characteristic of C-C bonds of aromatic ring [44].

### 3.2. Optimization of Extracting Parameters and Validation of the RSM Model

The experimental total phenolic content (TPC) found in the different extracts varied from $13.55 \pm 0.07$ mg GAE/100 g d.w. to $22.27 \pm 0.04$ mg GAE/100 g d.w. (runs 4 and 7, respectively; Table 1), in accordance with Zuorro et al. [33]. The lowest TPC value was found at a solvent composition equal to 80%, extraction time 60 min, and extraction temperature 50 °C. On the contrary, the highest TPC was achieved at 40% solvent composition, 40 min extraction time, and 60 °C extraction temperature.

Based on the experimental design given in Table 1, regression analysis was applied and a second-order polynomial model was developed. The mathematical relationship among the independent variables ($X_1$, $X_2$, $X_3$) and the response ($TPC_{RSM}$) were obtained as follows:

$$TPC_{RSM} = 53.03542 + 0.18485\,X_1 - 0.00290\,X_1^2 - 0.00273\,X_2 - 0.00199\,X_2^2$$
$$-1.60158\,X_3 + 0.01473\,X_3^2 - 0.00089\,X_1X_2 + 0.00137\,X_1X_3 + 0.00456\,X_2X_3$$

where $X_1$, $X_2$, and $X_3$ are the coded variables for solvent composition, extraction time, and extraction temperature, respectively. The magnitude and sign of the coefficients for intercept, linear, quadratic, and interaction effects point out the influence of each factor. The adequacy of the model was performed by the analysis of variance (ANOVA), as shown in Table 4. The linear, quadratic, and two-factor interaction effects coefficients; the statistical parameters F-values; the coefficient of determination $R^2$; the adjusted coefficient of determination $Rad_j^2$; and the lack of fit value are summarized in Table 4. The high F value for all responses indicated that the model obtained was statistically significant. The second-order polynomial model was highly significant ($p < 0.0001$), describing a high correlation degree between the experimental and predicted values, as shown by its coefficient of determination $R^2$ (0.999) and adjusted $Rad_j^2$ (0.998). Although the *p*-value of lack of fit was significant ($p = 0.0004$), the sum of square of lack of fit (0.0349) was less than the sum of square of pure error (0.0457). Therefore, this model can be accepted.

**Table 4.** Analysis of variance for the second-order polynomial equation for ultrasound-assisted extraction of total phenolic compounds (TPC).

| | Sum of Square | Degree of Freedom | Mean Square | F-Value | *p*-Value |
|---|---|---|---|---|---|
| $X_1$ | 163.7515 | 1 | 163.7515 | 71,071.23 | <0.0001 |
| $X_2$ | 1.5759 | 1 | 1.5759 | 683.99 | <0.0001 |
| $X_3$ | 44.1731 | 1 | 44.1731 | 19,171.94 | <0.0001 |
| $X_1^2$ | 14.9372 | 1 | 14.9372 | 6483.05 | <0.0001 |
| $X_2^2$ | 7.0229 | 1 | 7.0229 | 3048.08 | <0.0001 |
| $X_3^2$ | 24.0312 | 1 | 24.0312 | 10,429.99 | <0.0001 |
| $X_1 X_2$ | 1.5052 | 1 | 1.5052 | 653.29 | <0.0001 |
| $X_1 X_3$ | 0.8965 | 1 | 0.8965 | 389.11 | <0.0001 |
| $X_2 X_3$ | 9.9736 | 1 | 9.9736 | 4328.74 | <0.0001 |
| Residual | 0.0806 | 35 | 0.0023 | | |
| Lack of fit | 0.0349 | 3 | 0.01164 | 8.1400 | 0.0004 |
| Pure error | 0.0457 | 32 | 0.00143 | | |
| Total SS | 271.5927 | 44 | | | |
| $R^2$ | 0.999 | | | | |
| $R_{adj}^2$ | 0.998 | | | | |

Legend: $X_1$ = solvent composition, $X_2$ = extraction time, $X_3$ = extraction temperature.

While all terms were found to be significant ($p < 0.0001$), the linear term of the solvent composition ($X_1$) and, to a lesser extent, the linear and quadratic terms of the temperature ($X_3$) greatly affected the extraction yield of TPC from artichoke by-products. These results are in accordance with previous studies. In general, the yield of TPC increased with the increase in solvent composition from 0% to the intermediate level; after this, it gradually decreased [45,46]. The TPC increase at an intermediate level could suggest that the phenolic

compounds are more soluble in an ethanol/water solution than in a pure solvent. In particular, ethanol could increase the extraction yields and water could enhance swelling of cell material, increasing positively the contact of surface area between plant matrix and solvent, resulting in an increase in the extraction yield [47]. However, water and low concentration of ethanol can easily gain access to cells, but a high concentration of ethanol can cause protein denaturation, preventing the dissolution of polyphenols and, thus, influencing the extraction rate [47]. Moreover, the yield of TPC increased with the increase in temperature, which promoted the extraction by improving the rate of release of phytocompounds into the solvent, also in view of the higher dissociation of polyphenols linked with the membrane [45]. However, at temperatures higher than 70 °C, the extraction yield could decrease, probably due to the occurrence of degradative mechanisms such as oxidative processes and the heat-induced degradation of thermolabile compounds [35,45]. A longer extraction time eases the extraction of a high amount of phenolic compounds [35] as the wall cells were partially disrupted and the release of these compound is facilitated, especially at extraction temperatures around 60 °C for artichokes [47–49].

The effects on the overall response desirability of different combinations of levels of each pair of independent variables were evaluated using 3D-response surface plots (Figure 2). The desirability function showed the desirability of TPC (which can range from 0.0 for undesirable up to 1.0 for very desirable) across the observed range of each class. Therefore, the predicted optimal values for the independent variables to obtain the maximum yield of TPC were as follows: solvent composition ($X_1$) = 50%; extraction time ($X_2$) = 60 min; extraction temperature ($X_3$) = 60 °C. Under these optimal conditions, the extraction yield and the predicted yield of TPC were 22.4 ± 0.2 mg GAE/g d.w. and 22.5 mg GAE/g d.w., respectively.

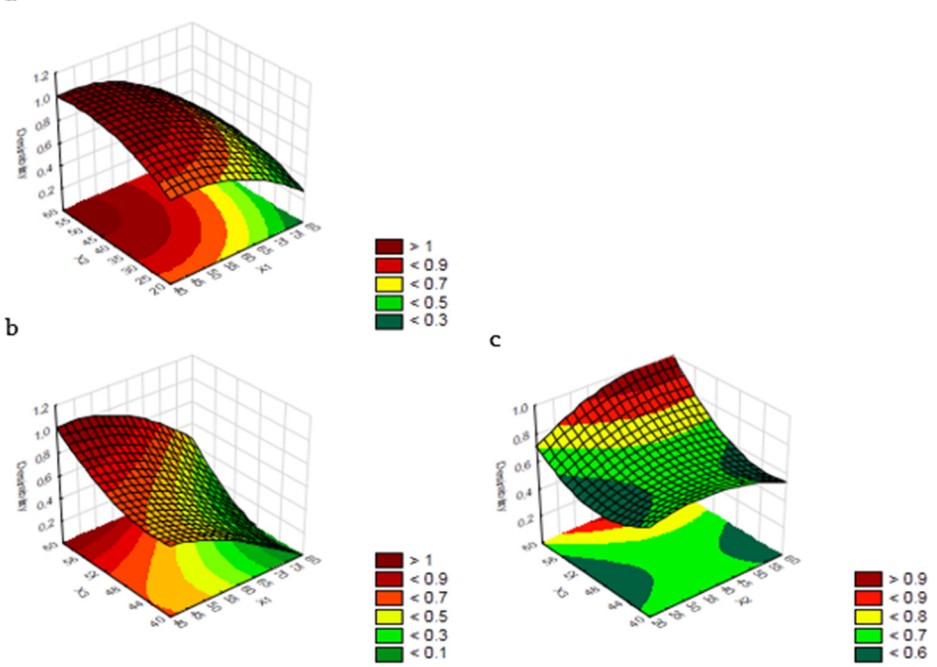

**Figure 2.** Response surface and contour plots of artichoke by-products extract: (**a**) desirability as a function of $X_1$ (solvent composition) and $X_2$ (extraction time); (**b**) desirability as a function of $X_1$ (solvent composition) and $X_3$ (extraction temperature); (**c**) desirability as a function of $X_2$ (extraction time) and $X_3$ (extraction temperature).

Comparing the results of the traditional extraction process with optimized UAE, an increase of 88% of TPC recovery was found. In fact, a TPC value of 11.9 ± 0.1 mg GAE g d.w. was obtained when extraction procedure without ultrasounds was applied. It can be due to the cavitation phenomenon induced at solid–liquid interface by UAE. As is well known,

cavitation facilitates the release of extractable compounds and enhances mass transport by disrupting the plant cell wall. Similar results were reported by Pasqualone et al. [29].

### 3.3. Phenolic Extract Characterization by UHPLC-ESI-QTOF MS

The composition of the phenolic substances in the optimized extract was determined by UHPLC-ESI-QTOF MS. Table 5 showed the identification and quantification of main substances. The phenolic compounds in *Cynara scolymus* species are potential sources of natural antioxidants for commercial purposes and are typically composed of hydroxycinnamic acids and flavonoids [50]. In particular, the method used was helpful to detect 51 phenolic compounds, of which 18 were identified (Table 5).

**Table 5.** Peak assignments of metabolites in optimized extract using UHPLC-Q-TOF MS in negative ionization mode, metabolites.

| Peak | Rt (min) | Compounds | Theoretical m/z | Experimental m/z | Molecular Formula | Error (ppm) | mg/g |
|---|---|---|---|---|---|---|---|
| 1 | 4.259 | Caffeoylquinic acid I * | 353.0878 | 353.0882 | $C_{16}H_{18}O_9$ | 1.13 | $0.77 \pm 0.04$ |
| 2 | 5.859 | 3-O-Caffeoylquinic acid (neochlorogenic acid) * | 353.0878 | 353.0884 | $C_{16}H_{18}O_9$ | 1.70 | $0.56 \pm 0.04$ |
| 3 | 12.492 | 5-O-Caffeoylquinic acid (chlorogenic acid) * | 353.0878 | 353.0885 | $C_{16}H_{18}O_9$ | 1.98 | $14.1 \pm 0.2$ |
| 4 | 13.346 | 4-O-Caffeoylquinic acid (cryptochlorogenic acid) * | 353.0878 | 353.0884 | $C_{16}H_{18}O_9$ | 1.70 | $0.67 \pm 0.05$ |
| 5 | 14.314 | Coumaroylquinic acid I ** | 337.0929 | 337.0937 | $C_{16}H_{18}O_8$ | 2.37 | $0.36 \pm 0.02$ |
| 6 | 14.747 | Dicaffeoylquinic acid I * | 515.1195 | 515.1204 | $C_{25}H_{24}O_{12}$ | 1.75 | $0.26 \pm 0.02$ |
| 7 | 14.864 | Caffeoylquinic acid dimer I * | 705.1672 | 705.1688 | $C_{32}H_{34}O_{18}$ | 2.27 | $0.18 \pm 0.01$ |
| 8 | 15.231 | Coumaroylquinic acid II ** | 337.0929 | 337.0935 | $C_{16}H_{18}O_8$ | 1.80 | $0.096 \pm 0.005$ |
| 9 | 16.831 | Quercetin 3-O-rutinoside (rutin) *** | 609.1461 | 609.1467 | $C_{27}H_{30}O_{16}$ | 0.98 | $0.038 \pm 0.003$ |
| 10 | 16.998 | Dicaffeoylquinic acid glucoside I * | 677.1723 | 677.1733 | $C_{31}H_{34}O_{17}$ | 1.48 | $0.29 \pm 0.03$ |
| 11 | 17.182 | Dicaffeoylquinic acid glucoside II * | 677.1723 | 677.1737 | $C_{31}H_{34}O_{17}$ | 2.07 | $0.67 \pm 0.05$ |
| 12 | 19.449 | Kaempferol-rutinoside *** | 593.1512 | 593.1518 | $C_{27}H_{30}O_{15}$ | 1.01 | $0.090 \pm 0.003$ |
| 13 | 19.683 | Dicaffeoylquinic acid II * | 515.1195 | 515.1206 | $C_{25}H_{24}O_{12}$ | 2.14 | $1.03 \pm 0.08$ |
| 14 | 20.066 | Dicaffeoylquinic acid III * | 515.1195 | 515.1195 | $C_{25}H_{24}O_{12}$ | 0.00 | $18.4 \pm 0.3$ |
| 15 | 20.383 | Dicaffeoylquinic acid IV * | 515.1195 | 515.1201 | $C_{25}H_{24}O_{12}$ | 1.16 | $14.4 \pm 0.3$ |
| 16 | 21.417 | Dicaffeoylquinic acid V * | 515.1195 | 515.1208 | $C_{25}H_{24}O_{12}$ | 2.52 | $3.5 \pm 0.1$ |
| 17 | 21.767 | Coumaroyl-caffeoylquinic acid * | 499.1246 | 499.1256 | $C_{25}H_{24}O_{11}$ | 2.00 | $0.47 \pm 0.04$ |
| 18 | 21.784 | Coumaroylquinic acid III ** | 337.0929 | 337.0941 | $C_{16}H_{18}O_8$ | 3.58 | $0.076 \pm 0.005$ |

\* equivalent of chlorogenic acids; ** equivalent of cumaric acids; *** equivalent of quercetin rutinoside.

All the compounds detected were uncertainly characterized by means of their detectable UV spectrum, MS data, and by comparison with those found in literature. Among the hydroxycinnamic acids, dicaffeoylquinic acid and chlorogenic acid were in greater concentration in the optimized extract, in agreement with Abu-Reidah et al. [51]. However, unlike Abu-Reidah et al. [51], we detected four isomers at different retention times (4.259, 5.859, 12.492, and 13.346 min) with the same [MH]ion at m/z 353.0878 and the same molecular formula ($C_{16}H_{17}O_9$).

These compounds have been assigned as mono-caffeoylquinic acid and its isomers [51,52]. Dicaffeoylquinic acid and its isomers, all characterized by precursor ion at m/z 515.1195 and identical molecular formula ($C_{25}H_{23}O_{12}$), were found at different retention times (14.747, 19.683, 20.066, 20.383, and 21.417 min). Among the flavonoids, several compounds belonging to the flavonol and flavone classes were found. Regarding flavonol derivatives, two compounds were detected at 16.831 and 19.449 min. Both compounds were suggested to be rutin isomers [51]. As reported in previous studies [46,50,52], different flavone derivatives were found in the extract. However, further studies are needed in order to identify them with certainty. The same authors found other compounds characteristic for *Cynara scolymus* species, such as the flavones Apigenin 7-glucuronide, Luteolin 7-rutinoside, Luteolin-7-O-glucoside, and Luteolin 7 glucuronide [46,50,52]. In addition, Pasqualone et al. [29] found significant differences in phenolic profiles among the three investigated cultivars and comparing US with untreated ultrasound extraction observed a non-univocal behavior of phenolic substances characterized by increases in some compounds whereas

other phenolics remains constant or even decreased probably due to condensation and polymerization phenomena.

### 3.4. Effect of Artichoke By-Products Optimal Extract on Fresh Egg Pasta

The appearance and the color parameters for artichoke by-products (ABP), and uncooked ($UP_i$) and cooked ($CP_i$) fresh egg pasta are shown in Figure 3 and Table 6.

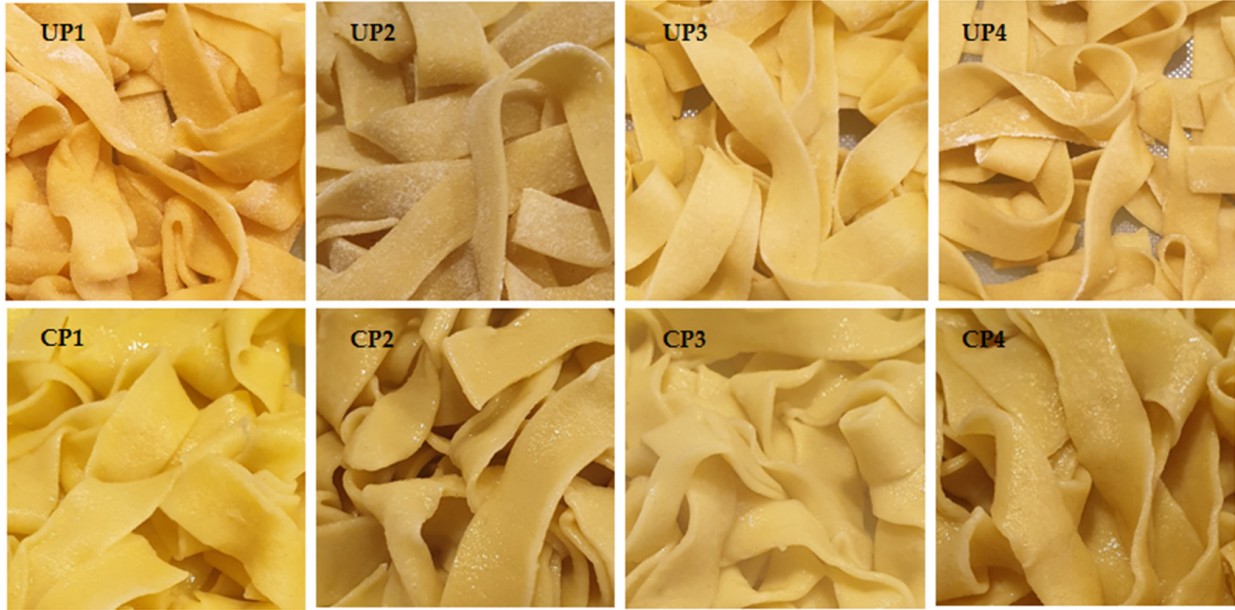

**Figure 3.** Appearance of uncooked (UPi) and cooked (CPi) pasta samples.

**Table 6.** Color characteristics (L* = luminosity, a* = redness, b* = yellowness) for artichoke by-products (ABP), and uncooked ($UP_i$) and cooked ($CP_i$) fresh egg pasta.

|  | L* | a* | b* |
|---|---|---|---|
| ABP | 61.50 ± 0.62 e | −0.17 ± 0.15 a | 17.76 ± 0.05 f |
| $UP_1$ | 87.90 ± 0.66 b | −2.30 ± 0.18 e | 26.84 ± 1.04 bc |
| $UP_2$ | 88.55 ± 0.41 ab | −2.11 ± 0.03 e | 24.50 ± 0.92 cd |
| $UP_3$ | 89.25 ± 0.48 a | −1.20 ± 0.04 b | 23.00 ± 1.22 de |
| $UP_4$ | 88.58 ± 0.19 ab | −1.26 ± 0.05 b | 22.08 ± 0.69 e |
| $CP_1$ | 87.15 ± 0.39 b | −3.28 ± 0.09 g | 31.86 ± 0.87 a |
| $CP_2$ | 80.37 ± 0.89 d | −1.97 ± 0.06 d | 25.77 ± 0.78 c |
| $CP_3$ | 86.91 ± 0.29 b | −2.77 ± 0.08 f | 27.85 ± 0.31 b |
| $CP_4$ | 83.19 ± 0.31 c | −1.79 ± 0.08 c | 23.79 ± 0.41 d |

Different letters in each column indicate significant difference ($p < 0.05$).

The visual observation of Fettuccine addressed a good pasta quality for all formulation. The enrichment of ABP and/or the presence of semolina caused a decrease, more or less pronounced, in luminosity (L*) for all pastas. Regarding redness, the addition of ABP does not significantly modify the samples of uncooked pasta. On the contrary, the significant differences between $UP_1$-$UP_2$ and $UP_3$-$UP_4$ could be due to the presence of the semolina. The cooked pasta samples showed significantly higher greenness values, especially for $CP_1$ compared with uncooked samples. Pasta with semolina showed higher values of yellowness, probably due to a larger amount of semolina's carotenoids. At last, all cooked pasta samples presented higher values of yellowness than the uncooked ones. The increase in parameters a* and b* was related to the high content of carotenoids in egg yolk, and adding this to pasta making increased the yellowness and redness of the pasta at each time. A similar increase in yellowness in the case of the cooked pasta was also reported by Teterycz et al. [53].

Differences in CIELab coordinates among all cooked samples could be due to a variation in the microstructure of the fresh pasta matrix after cooking and to a loss of water-soluble compounds during cooking.

Cooking can influence the concentration of bioactives in the pasta [8]. Ensuring that their bioavailability is preserved after processing and cooking phases is crucial for commercializing a functional product [26]. In fact, pasta cooking might cause significant loss of bioactive compounds, such as polyphenols, due to their release in the cooking water or their degradation at high temperatures [8,54,55]. Therefore, as suggested by Oliviero and Fogliano [8], in order to reduce the losses of phytochemicals into the cooking water, boiling was achieved with little water. Table 7 shows the TPC concentration in different formulation of pasta, with or without extract of artichoke by-products. As expected, the two pasta formulations with artichoke by-products extract ($P_2$ and $P_4$) showed higher polyphenol content than the two control ones ($P_1$ and $P_3$). A strong decrease in TPC was observed as a result of the cooking process, especially for the two control formulations ($-71\%$ for $P_1$ and $-70\%$ for $P_3$). On the contrary, the reduction of TPC in the cooked product is relatively low for $P_4$ ($-55\%$), in comparison with $P_2$ ($-64\%$). These results could be related to the different formation of the gluten network during kneading for the four types of dough, whose raw materials have different physical and chemical characteristics. In fact, the gluten quality is generally affected by the proteins' aminoacidic composition and ratio of amylose and amylopectin in starch, which are strongly dependent on the species, the genotype, environmental condition, and agronomic practices. Moreover, as reported by Palermo et al. [56], cooking can induce many chemical and physical changes in foods, among which are a reduction in polyphenols content due to the thermal degradation and their solubilization in cooking water. The impact of cooking on their concentration depends on the processing parameters, the structure of food matrix, and the chemical nature of the compound [56].

**Table 7.** Total phenolic content (TPC) for uncooked ($UP_i$) and cooked ($CP_i$) fresh egg pasta and relative percentage change in TPC ($\Delta$TPC) before and after cooking.

| | TPC-$UP_i$ (mg GAE/g d.w.) | TPC-$CP_i$ (mg GAE/g d.w.) | $\Delta$TPC (%) |
|---|---|---|---|
| $P_1$ | 1.86 ± 0.03 c | 0.54 ± 0.02 c | −71 |
| $P_2$ | 2.05 ± 0.05 a | 0.73 ± 0.04 b | −64 |
| $P_3$ | 1.92 ± 0.03 b | 0.57 ± 0.03 c | −70 |
| $P_4$ | 2.04 ± 0.02 a | 0.91 ± 0.02 a | −55 |

According to Zardetto et al. [39], fresh pasta stored at temperature of 5 °C could show mold colonies of 3 mm in diameter in approximately 6–7 days. Our study demonstrated that mold colonies appeared after 6 days for $P_1$ and $P_3$, after 7 days for $P_2$, and after 8 days for $P_4$ (Figure 4).

The results of the GzLM analysis (Table 8) showed that the two factors were highly significant ($\chi^2$ =22.322, $p < 0.0001$ for P; and $\chi^2$ =14.567, $p = 0.0010$ for day). The two-way interaction was not considered because it resulted as not significant. GzLM also highlighted significant differences among control pastas and enriched pastas ($p < 0.0001$), and a slight difference between $P_2$ and $P_4$ ($p = 0.0380$).

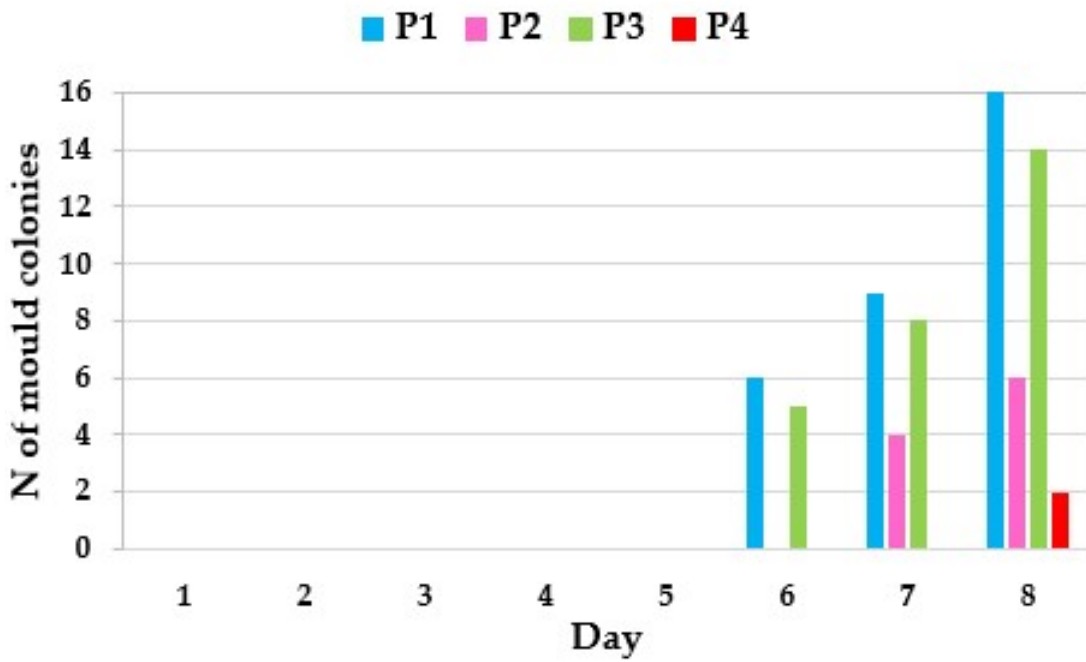

**Figure 4.** Number of mold colonies that appeared on cooked pasta samples for 8 days of refrigerator storage.

**Table 8.** Generalized linear model for the response variable (number of mold colonies) as a function of the type of pasta (P) and day.

| Factor | $X^2$ | $Pr > X^2$ |
|---|---|---|
| P | 22.322 | <0.0001 |
| Day | 14.567 | 0.0010 |

These results supported the hypothesis that the natural extract with high antioxidant power could positively influence the maintenance of the product, elongating its shelf life. Moreover, they appeared to be of great importance since the literature highlighted that fresh filled pasta packaged in ordinary atmosphere and stored at 4 °C generally showed very short shelf life values, accounting for hours to one week, depending on the hygienic production conditions [57].

## 4. Conclusions

Our study showed that artichoke by-products are rich in phenolic compounds, especially in hydroxycinnamic acids, which can be easily extracted using a green chemistry approach. UAE has proven to be an effective technique for their sustainable recovery, setting the optimal extraction conditions at 60 °C, 60 min, and using a mixture of 50% ethanol as solvent. These extracts are a viable source of bioactives with a positive activity that can be used to design sustainable food functional products, as our new functional pasta. Moreover, our preliminary results demonstrated that the enrichment of Fettuccine with artichoke by-products extract reduced the losses of polyphenols into the cooking water and increased the rejection time of the pasta product, assessing a longer shelf life and lower spoilage, especially for soft wheat pasta formulation. Based on these encouraging findings, future research will have to go more in depth on the effect of the extract addition on technological and sensorial properties of the new pasta and focus on the study of the in vivo bioavailability of these functional ingredients, in order to support possible upscaling to an industrial level and to demonstrate that it can represent a sustainable means to increase antioxidant dietary intake.

**Author Contributions:** Conceptualization, T.A. and R.C. (Roberto Ciccoritti); methodology, T.A. and R.C. (Roberto Ciccoritti); software, T.A.; validation, T.A., R.C. (Roberto Ciccoritti) and D.C.; formal analysis, F.M., R.C. (Roberto Ciorba) and R.C. (Roberto Ciccoritti); investigation, T.A., R.C. (Roberto Ciccoritti), D.C., and S.R.; data curation, T.A., R.C. (Roberto Ciccoritti) and D.C.; writing—original draft preparation, T.A., R.C. (Roberto Ciccoritti) and S.R.; writing—review and editing, T.A., R.C. (Roberto Ciccoritti), R.C. (Roberto Ciorba), D.C., and S.R. All authors have read and agreed to the published version of the manuscript.

**Funding:** This research received no external funding.

**Institutional Review Board Statement:** Not applicable.

**Informed Consent Statement:** Not applicable.

**Data Availability Statement:** The data presented in this study are available on request from the corresponding author.

**Acknowledgments:** Grateful acknowledgements are given to Luigi Bartoli and Paolo Gabrielli for technical assistance.

**Conflicts of Interest:** The authors declare no conflict of interest.

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
