# Peer review of "Artichoke By-Products Valorization for Phenols-Enriched Fresh Egg Pasta: A Sustainable Food Design Project"

_sustainability, doi:10.3390/su142214778_

Round 1

Reviewer 1 Report

Comments to the Authors

The manuscript “Artichoke by-products valorization for phenols-enriched fresh egg pasta: a sustainable food design project” has been well written, the experimental set up well designed and the results provided are relevant to optimize the US-assisted  extraction of phenolic compounds from artichoke by-products. However, the novelty of the manuscript and its contribution to the advancement of the current state of the art should be better highlighted, as well as the language of the paper needs to be thoroughly checked.

Here are my detailed comments:

Abstract and keywords

- Lines 20-21: reduce the significant digits or at least uniform them to the rest of the manuscript (e.g. in Line 19 fewer digits are reported for TPC). Likewise, express the TPC and the concentrations of the individual phenolic compounds identified with the same units of measurement (both in mg/g dw or in mg/kg dw).

- Line 28: before reporting the results related to the shelf life, clearly explicit in few words in the abstract that, apart from the experiments and analysis already highlighted, microbiological analyses on the pasta shelf life have also been carried out.

- Line 30: replace “extract” with “extracted”.

- Keywords: replace “biowastes” with “agri-food residues” or “by-products”, and do it throughout all the manuscript. Replace “Surface response methodology” with “ Response Surface Methodology”.

 Introduction

- Lines 41-44: rephrase this sentence, it is too long and hard to understand.

- Line 55: add “sources” at the end of the statement to the adjective “promising”.

- Lines 81-81 replace: “An economically viable solution for the valorization of these by-products is their recovery for pharmaceutical or cosmetic purposes”.

- Lines 90-93: better explain why did you choose Ultrasounds among all the proposed emerging technologies used to enhance the extractability of phenolic compounds from agri-food by-products (for example Pulsed Electric Fields as well as Microwaves have been efficiently applied on artichoke external bracts and stems to recover phenolic compounds).

- Line 96: explicit which kind of by-products did you focus on (e.g. bracts, stems)

- Lines 98-101: in my opinion, it is better to move this part before line 94. The sense of the manuscript will flow better highlighting first the shortcomings emerging from the literature and then how the authors want to fill these lacks or contribute to their advancement.

- Line 100: No literature data on artichoke by-products with this purpose.

Actually, there are literature data on this topic. There is a previous study (Pasqualone, A., Punzi, R., Trani, A., Summo, C., Paradiso, V. M., Caponio, F., Gambacorta, G. Enrichment of fresh pasta with antioxidant extracts obtained from artichoke canning by-products by ultrasound-assisted technology and quality characterisation of the end product. 2017, Food science and technology. doi:10.1111/ijfs.13486), not included in this manuscript, that explored the feasibility of employing artichoke by-product extracts obtained by ultrasound-assisted extraction for the production of functional fresh pasta by determining the phenolic profile, colour, textural properties and cooking performances.

Therefore, I suggest clearly highlighting the novelty of this paper compared to the scientific material still available on the same topic in order to give a comprehensive overview of how this work will contribute to an advance of the knowledge about a specific issue.

Materials and Methods

- Line 111: explicit which artichoke by-products did you focus on (bracts, stems, or a mixture of them).

- Lines 119-120: did you take the protein content from the proximate composition reported on the package of the products or did you measure it? If yes, add the methodology used.

- Lines 141-142: How did you choose the best S/L ratio (2/40 (g/mL))?

- Line 143: apart from the frequency of US apparatus, what is its power, is it fixed or adjustable?

- Line 145: What is the reason behind the selection of Box-Behnken design?

- Lines 154-155: demonstrated “that” higher extraction times and temperatures increased “the” yields only “of” about 5% and were not economically sustainable. Add the words in “”.

- Line 162: the S/L ratio 20:80 (v/v) seems to be not coherent with the one reported in Lines 141-142. Why is it expressed in v/v?

- Line 165: what is the pore size of the nylon syringe filter?

- Line 178: did you completely dehydrate the extract solution? Did you check the absence of ethanol or its concentration has only been reduced?

- Line 195: ‘The injection volume was 1µL’. Explain if the extract has been diluted and how it was prepared before injection.

- Line 206: uniform the units of measurement throughout the manuscript. It is better to express all in mg/g dw.

- Line 209: using five concentrations spanning 10, 25, 50, 100 ppm. They are four concentrations.

- Line 223 and Line 229: the addition of a plant extract, and of semolina in the recipe of pasta affects the mixing time as well as the optimal cooking time of the pasta, that should be optimized and tailored to the addition of the different ingredients and the different compositions investigated. Did you check how and if the different compositions affected these two parameters?

-Line 244: “observed daily”, explicit for how many days did you investigate the shelf life of the fresh pasta.

Results and discussion

- Line 273: remove the dot after cellulose. Add “of” after range.

- Where is the control? I mean, the artichoke by-products without US treatment. Did you not compare the US-assisted extraction with the conventional extraction? It is advisable to see if the application of US is able to reduce the diffusion time, the extraction temperature, or the ethanol concentration with respect to the control samples in order to achieve the same extraction yields. Likewise, it would be interesting to see if applying the same extraction conditions, the US could improve the extraction yields compared to the control.

- Line 298: Table 4 capital letter.

- Line 303: high degree of correlation or high correlation degree.

- Line 306: report in brackets the values of the sum of squares.

- Lines 315-316: discuss more in detail the reasons why the TPC values usually tend to increase up to reach a maximum value and decrease for higher ethanol concentrations.

Lines 376: explain what this phenomenon might be due to.

Line 424: “50%” of water, is not necessary.

Line 432: move in vivo before bioavailability.

Line 434: a sustainable mean, not means.

Author Response

Comments to the Authors

The manuscript “Artichoke by-products valorization for phenols-enriched fresh egg pasta: a sustainable food design project” has been well written, the experimental set up well designed and the results provided are relevant to optimize the US-assisted extraction of phenolic compounds from artichoke by-products. However, the novelty of the manuscript and its contribution to the advancement of the current state of the art should be better highlighted, as well as the language of the paper needs to be thoroughly checked.

Thanks for your kind comments. We are grateful for the reviewer’s constructive suggestions. We have addressed all the issues point-by-point in our response below

Abstract and keywords

- Lines 20-21: reduce the significant digits or at least uniform them to the rest of the manuscript (e.g. in Line 19 fewer digits are reported for TPC). Likewise, express the TPC and the concentrations of the individual phenolic compounds identified with the same units of measurement (both in mg/g dw or in mg/kg dw).

We have done it.

- Line 28: before reporting the results related to the shelf life, clearly explicit in few words in the abstract that, apart from the experiments and analysis already highlighted, microbiological analyses on the pasta shelf life have also been carried out.

We modified the text. We did not conduct microbiological analyzes in this work, but we relied only on image analysis. One of the purposes of the study was to understand if the antimicrobial power of the artichoke described in the literature could have effects on the shelf life of the pasta. In a future study we will deepen the question from a microbiological point of view, optimizing the use of the extract to have a maximum shelf life.

- Line 30: replace “extract” with “extracted”.

We have done it.

- Keywords: replace “biowastes” with “agri-food residues” or “by-products”, and do it throughout all the manuscript. Replace “Surface response methodology” with “ Response Surface Methodology”.

We have done it.

Introduction

- Lines 41-44: rephrase this sentence, it is too long and hard to understand.

We have done it.

- Line 55: add “sources” at the end of the statement to the adjective “promising”.

We have done it.

- Lines 81-81 replace: “An economically viable solution for the valorization of these by-products is their recovery for pharmaceutical or cosmetic purposes”.

We have done it.

- Lines 90-93: better explain why did you choose Ultrasounds among all the proposed emerging technologies used to enhance the extractability of phenolic compounds from agri-food by-products (for example Pulsed Electric Fields as well as Microwaves have been efficiently applied on artichoke external bracts and stems to recover phenolic compounds).

We added a sentence to explain the choice of UAE.

- Line 96: explicit which kind of by-products did you focus on (e.g. bracts, stems)

We have done it.

- Lines 98-101: in my opinion, it is better to move this part before line 94. The sense of the manuscript will flow better highlighting first the shortcomings emerging from the literature and then how the authors want to fill these lacks or contribute to their advancement.

We have done it.

- Line 100: No literature data on artichoke by-products with this purpose.

Actually, there are literature data on this topic. There is a previous study (Pasqualone, A., Punzi, R., Trani, A., Summo, C., Paradiso, V. M., Caponio, F., Gambacorta, G. Enrichment of fresh pasta with antioxidant extracts obtained from artichoke canning by-products by ultrasound-assisted technology and quality characterisation of the end product. 2017, Food science and technology. doi:10.1111/ijfs.13486), not included in this manuscript, that explored the feasibility of employing artichoke by-product extracts obtained by ultrasound-assisted extraction for the production of functional fresh pasta by determining the phenolic profile, colour, textural properties and cooking performances.

Therefore, I suggest clearly highlighting the novelty of this paper compared to the scientific material still available on the same topic in order to give a comprehensive overview of how this work will contribute to an advance of the knowledge about a specific issue.

Thank you for reporting this paper to us. We explained the differences of our study with the paper by Pasqualone et al.

Materials and Methods

- Line 111: explicit which artichoke by-products did you focus on (bracts, stems, or a mixture of them).

We have done it.

- Lines 119-120: did you take the protein content from the proximate composition reported on the package of the products or did you measure it? If yes, add the methodology used.

The values of the protein content for soft wheat flour, semolina and eggs are those reported on the package of the products. We specified it in the text

- Lines 141-142: How did you choose the best S/L ratio (2/40 (g/mL))?

From literature data the phenolic content of artichoke by-products showed high variability in relation to the histological tissue, varieties used, maturation stage, etc., and different ratio S/L were reported (usually 1:3, 1:5, 1:10) to extract bioactives. The choice of different ratio should be related to the chemical nature of the matrix. However, high variability of chemical composition / compound extractability was found in artichoke by-products due to histological tissue, cultivars, maturation stage etc. Therefore, it should be evaluated case by case. For these reasons, we do not consider S/L ratio among the parameter to optimize. The choice to use the S/L ratio 1:20 (higher than what on average is present in the literature) was used to exclude a solvent saturation during the extraction.

- Line 143: apart from the frequency of US apparatus, what is its power, is it fixed or adjustable?

The power is fixed

- Line 145: What is the reason behind the selection of Box-Behnken design?

We added a sentence to explain the choice of BBD.

- Lines 154-155: demonstrated “that” higher extraction times and temperatures increased “the” yields only “of” about 5% and were not economically sustainable. Add the words in “”.

We added the suggested words

- Line 162: the S/L ratio 20:80 (v/v) seems to be not coherent with the one reported in Lines 141-142. Why is it expressed in v/v?

The ratio reported at line 141-142 is referred as extract artichoke powdered gram in solvent. At line 162 the different ratio is referred by the solvent composition (water: ethanol) which is measured in volume.

- Line 165: what is the pore size of the nylon syringe filter?

Syringe filter pore was added

- Line 178: did you completely dehydrate the extract solution? Did you check the absence of ethanol or its concentration has only been reduced?

The extract solution was constituted by ethanol /water solution and was subject to rotovapor at 40°C to reduce the alcoholic concentration. For this reason, the sentence was changed

- Line 195: ‘The injection volume was 1µL’. Explain if the extract has been diluted and how it was prepared before injection.

The explanation was added

- Line 206: uniform the units of measurement throughout the manuscript. It is better to express all in mg/g dw.

We modified units of measurement in all text

- Line 209: using five concentrations spanning 10, 25, 50, 100 ppm. They are four concentrations.

We are sorry for the mistake. We provided to correct it

- Line 223 and Line 229: the addition of a plant extract, and of semolina in the recipe of pasta affects the mixing time as well as the optimal cooking time of the pasta, that should be optimized and tailored to the addition of the different ingredients and the different compositions investigated. Did you check how and if the different compositions affected these two parameters?

In order to decide the mixing and cooking times for the two recipes, we tried different recipes with different quantities of the various ingredients, and we chose those that allowed comparable times.

-Line 244: “observed daily”, explicit for how many days did you investigate the shelf life of the fresh pasta.

We added days of investigation

Results and discussion

- Line 273: remove the dot after cellulose. Add “of” after range.

We have done it.

- Where is the control? I mean, the artichoke by-products without US treatment. Did you not compare the US-assisted extraction with the conventional extraction? It is advisable to see if the application of US is able to reduce the diffusion time, the extraction temperature, or the ethanol concentration with respect to the control samples in order to achieve the same extraction yields. Likewise, it would be interesting to see if applying the same extraction conditions, the US could improve the extraction yields compared to the control.

Thank you for the suggestion. Methods, results and discussion about control was added

- Line 298: Table 4 capital letter.

We did not understand. Table 4 is already in capital letter

- Line 303: high degree of correlation or high correlation degree.

We have done it.

- Line 306: report in brackets the values of the sum of squares.

We have done it.

- Lines 315-316: discuss more in detail the reasons why the TPC values usually tend to increase up to reach a maximum value and decrease for higher ethanol concentrations.

The discussion was added

Lines 376: explain what this phenomenon might be due to.

An explanation was added

Line 424: “50%” of water, is not necessary.

We have done it.

Line 432: move in vivo before bioavailability.

We have done it.

Line 434: a sustainable mean, not means.

We have done it.

Reviewer 2 Report

In this manuscript, the authors used UAE to obtain the artichoke extract rich in TPC, and then used the artichoke extract to prepare fresh egg pasta. The extraction method was optimized, the profile of TPC was investigated and the quality of the pasta was evaluated. The results are interesting. However, there are still issues should be addressed:

1. TPC can be easily oxidized during processing and storage which may cause deterioration for the pasta, for example darkening of the pasta. How to avoid?

2.  The authors found that "mould colonies appeared after 6 days for P1 and P3, after 7 days for P2, and after 8 days for P4". If you want to conclude that "high antioxidant power could positively influence the maintenance of the product, elongating its shelf life" more accurate assessment and statistical analysis should be carried out.

Author Response

In this manuscript, the authors used UAE to obtain the artichoke extract rich in TPC, and then used the artichoke extract to prepare fresh egg pasta. The extraction method was optimized, the profile of TPC was investigated and the quality of the pasta was evaluated. The results are interesting. However, there are still issues should be addressed

Thanks for your kind comments. We are grateful for the reviewer’s constructive suggestions. We have addressed all the issues point-by-point in our response below

  1. TPC can be easily oxidized during processing and storage which may cause deterioration for the pasta, for example darkening of the pasta. How to avoid?

We agreed to the reviewer as regards the easy oxidation of phenolic compounds. However, conflicted results have been reported with regard to the effect of pasta making process (mixing, extrusion, and drying) on phenolic acids in pasta. Verardo et al. (2011) found that the main effect of pasta processing was a decrease of free phenolic acids. Similarly, Fares et al. (2010) observed a decrease in free phenolic content of durum wheat pasta enriched with bran fractions of wheat, but the bound phenolics did not change. On the other hand, the results from the study of Khan, Yousif, Johnson, and Gamlath (2013) contradict those of the two mentioned studies; the pasta processing did not change the free and bound phenolic acid content of fettuccine made by replacing durum wheat semolina with red or white sorghum flour. The discrepancy between the mentioned studies could be explained by different conditions applied by the authors. The severity and mode of thermal process could also have a bearing on the phenols in the food. For these reasons, different precautions can and must be adopted to slow down the oxidative process. In particular, during the adopted transformation process artichoke by-products was stabilized using low temperature (the samples were immediately frozen). UAE was made at temperatures lower than 70°C. As reported in literature, this threshold is important because below which the alteration of bioactive compounds is reduced. The ethanol evaporation was made at 40°C and the extract immediately added to the ingredient to develop dough. In addition, during pasta making, to limitate the phenols oxidation we adopted low temperatures during all fases. Although it was not possible to completely exclude the oxidation process, the final product was equally higher in TPC as also reported in a previously work by Pasqualone et al 2017. Moreover, different reaction could be developed during pasta process that are responsible of pasta darkening (i.e. Maillard, etc.) and further study are need to evaluate the effective cause of the colour variation.

  1. The authors found that "mould colonies appeared after 6 days for P1 and P3, after 7 days for P2, and after 8 days for P4". If you want to conclude that "high antioxidant power could positively influence the maintenance of the product, elongating its shelf life" more accurate assessment and statistical analysis should be carried out.

Statistical analysis (Generalized linear model) was added

Round 2

Reviewer 1 Report

Comments to the Authors

- Line 152: Apart from the frequency, add also the power of the used US equipment.

- Line 173: Sample-to-solvent ratio 20:80 (v/v) was fixed in this study.

If 20:80 (v/v) is referred to the solvent composition (water: ethanol) substitute “Sample-to-solvent ratio” with “Water-ethanol mixture (20:80 (v/v))”

- Line 327: table 4 is written in lowercase letter. Change it in Table 4.

- Figure 2: better describe what each of the three graphs represents in the caption of Figure 2, maybe adding some letters a, b and c to identify the graphs.

- Line 430: substitute “that adding” to “and adding”.

Author Response

Thanks again for the suggestions.
all suggestions have been entered.

Reviewer 2 Report

I accept the revisions.

Author Response

Thanks again for the suggestions